

# Comparison of gut microbiota between adults with autism spectrum disorder and obese adults

Qiang Zhang[1,2], Rong Zou[2], Min Guo[2], Mengmeng Duan[2], Quan Li[1] and Huajun Zheng[2]

[1] Department of Obstetrics and Gynecology, Affiliated Hospital of Zunyi Medical University, Zunyi, China
[2] NHC Key Lab. of Reproduction Regulation (Shanghai Institute of Planned Parenthood Research), Fudan University, Shanghai, China

## ABSTRACT

**Background**. Autism spectrum disorder (ASD) and obesity are serious global public health problems. Studies have shown that ASD children are at a higher risk of obesity than the general population. To investigate the gut microbe characteristics of adults ASD and obese adults, we compared the gut microbiota of adults with ASD to obese adults.

**Methods**. The fecal samples were collected from 21 adult patients with ASD and 21 obese adults, and V3–V4 regions of 16S rRNA genes were sequenced by high-throughput DNA sequencing. The gut microbiota of adults with ASD and obese adults was compared.

**Results**. We observed the proportion of *Firmicutes/Bacteroidetes* in ASD was significantly increased, with families *Lachnospiraceae* and *Ruminococcaceae* significantly enriched in adult ASD. Eighteen genera, including *Lachnospiracea incertae sedis*, *Ruminococcus*, *Blautia*, and *Holdemanella* were significantly increased in adult ASD, whereas *Megamonas* and *Fusobacterium* were significantly increased in obesity. At the species level, we found six species enriched in ASD and three species enriched in obesity, including *Phascolarctobacterium succinatuten* producing propionate. *Dialister succinatiphilus* may be as a biomarker for predicting obesity, as well as *Prevotella copri* may be a common-owned pathogens of ASD and obesity.

**Conclusions**. Some conflicting results have been reported in microbiota studies of ASD, which may be related to age and obesity. Thus, the body mass index should be evaluated before analyzing the gut microbiota of patients with ASD, as obesity is prevalent in these individuals and gut microbiota is severally affected by obesity.

Corresponding authors
Quan Li, 1013283493@qq.com
Huajun Zheng, zhenghj@chgc.sh.cn

## INTRODUCTION

Autism spectrum disorder (ASD) is a group of complex neurodevelopmental disorders characterized by persistent deficits in social reciprocity and verbal/nonverbal social interaction communicative behaviors, as well as the presence of repetitive and restricted patterns of behaviors, activities, or interests (*American Psychiatric Association, 2013*).

According to the report of World Health Organization (WHO) (https://www.who.int/news-room/fact-sheets/detail/autism-spectrum-disorders), ASD begins before the age of 3 and persists throughout a person's life, with an average of one in 160 children worldwide suffering from ASD. There are great differences in intelligence of individuals with ASD (*American Psychiatric Association, 2013*), and most of them need lifelong care from family and society, which greatly influence patients' physical and mental health as well as socio-economic development. Therefore, ASD is a serious global public health problem.

Accumulating evidence has indicated that ASD children are at a higher risk of obesity than the general population (*Dhaliwal et al., 2019*; *Healy, Aigner & Haegele, 2019*; *Hill, Zuckerman & Fombonne, 2015*; *Zheng et al., 2017*), and autistic adults were more likely to be overweight or obese than non-autistic people (*Sedgewick, Leppanen & Tchanturia, 2020*). Obesity is a complex metabolic disease with unclear etiology, which usually defined as according to body mass index (BMI). In adults, obesity is defined as a BMI of $\geq 30$ kg/m$^2$ (*Bray et al., 2018*; *Jensen et al., 2014*). According to the report of WHO (https://www.who.int/zh/news-room/fact-sheets/detail/obesity-and-overweight), in 2016, more than 650 million adults were obese. With increasing prevalence of obesity, the risk of obesity-associated diseases such as cardiovascular disease, stroke, type 2 diabetes, hypertension, non-alcoholic fatty liver disease, and some types of cancer is increasing (*Jensen et al., 2014*; *Ogden et al., 2014*). Thus, obesity is also a serious public health problem.

In recent decades, increasing attention has been paid to the role of intestinal microbiota on both health and disease. Harmonious symbiosis of intestinal microbiota is the key to maintaining human health. Once the micro-ecological balance is broken, which probably lead to variety disorders, including ASD (*Fattorusso et al., 2019*) and obesity (*Mitev & Taleski, 2019*). Several studies have suggested that the microbiota-gut-brain axis plays a vital role in the occurrence and development of ASD (*Luna, Savidge & Williams, 2016*; *Martin et al., 2018*; *Van Sadelhoff et al., 2019*). The brain can affect the composition of gut microbiota through regulating host intestinal motility, secretion and permeability, and then bring about gastrointestinal symptoms of individuals with ASD (*Luna, Savidge & Williams, 2016*; *Martin et al., 2018*; *Van Sadelhoff et al., 2019*). Moreover, gut microbiota in turn affects the function of central nervous system (CNS) in the host via neurotransmitter, immune, or metabolite products, which can lead to the ASD-like behaviors (*MacFabe, 2015*; *Van Sadelhoff et al., 2019*).

Additionally, another study has shown that the microbiota-gut-brain axis also plays an important role in the development of obesity (*Torres-Fuentes et al., 2017*). Intestinal microbiota may be contributed to the occurrence and development of obesity by influencing the host's nutrient metabolism, energy balance, inflammation, and insulin resistance (*Khan et al., 2016*; *Torres-Fuentes et al., 2017*). Moreover, intestinal microbiota and its metabolites can directly stimulate the vagus and transmit stimulus signals to the CNS or indirectly act on the CNS through immune-neuroendocrine mechanisms, then affecting the feeding behavior of the body (*Fetissov, 2017*; *Torres-Fuentes et al., 2017*). In addition, the CNS can control the feeding behavior of host, and then provide nutrition for intestinal flora or affect the composition of intestinal microbiota (*Fetissov, 2017*).

Through association between obesity and ASD is often reported, most studies focused on risk factors contributing to obesity, like individuals with ASD often have picking eating behavior, spend less time on physical activities, have comorbidities associated with obesity, etc (*Dhaliwal et al., 2019*; *Zheng et al., 2017*). But how does obesity affect ASD has not been reported. *Stanislawski et al., (2019)* reported that a lower alpha diversity and a higher relative abundance of *Prevotella* are positively correlated with obesity among black and Hispanic populations. In our study, we also observed that the *Prevotella* was significantly increased in adults with ASD compared to healthy adults (in press). So we postulate that gut microbiota changes caused by obesity might be a contributing factor affecting ASD development. But comparison of intestinal microbiota characteristics between patients with ASD and obese patients has not been reported. Therefore, in this study, we determined and compared gut microbiota of 21 adult patients with ASD and 21 obese adults, to identify the similarities and differences of intestinal microbiota between them. Based on this, we expect to provide potential therapies and preventive measures for patients with ASD or obesity.

## MATERIALS AND METHODS

### Sample collection

Twenty-one patients diagnosed with ASD (mean BMI = 22.8, 15.9–31.9, with 6 females and 15 males) with ages ranging from 17 to 32 were recruited from the XinWangAi Caring Center for People with Intellectual Disability (Jinan, Shandong Province, China), and their care costs are mainly from social donation and government financial expenditure. These patients with ASD are all Han nationality, and diagnosed in childhood by clinicians according to the diagnostic criteria for childhood autism in International Classification of Diseases and Related Health Problems, 10th Revision (ICD-10) (*World Health Organization, 1993*). Patients with schizophrenia or other psychosis, or having taken antibiotics for one month prior to fecal sample collection were excluded. Most of the 21 ASD patients have gastrointestinal symptoms such as constipation and diarrhea. Twenty-one gender and age matched obese adults (mean BMI = 35.3, 31.4–49.6) that did not suffer from ASD, other neurodevelopmental disorders or neuropsychiatric diseases, and were not under dietary or medication control to lose weight, were recruited from a gym in Jinan. Stool specimens were collected during the daytime using MicroLocker[T] stool sample collector (YM-F02B, JiangSu YIMI Biotech Inc., China) which contains fecal sample preservation solution, and transferred to laboratory within three hours . All samples were stored at −80 °C until DNA extraction. The study was approved by the Medical Ethical Committee of Shanghai Institute of Planned Parenthood Research (NO: PJ2019-17). Written informed consent was obtained from the parents/guardians for all participants involved in this study. All methods were performed in accordance with the Declaration of Helsinki.

### Genomic DNA extraction, PCR amplification, and 16S rRNA gene sequencing

DNA extraction and PCR amplification were performed as described previously (*Zou et al., 2020*). Specially, the fecal DNA was extracted using the QIAamp DNA Stool Mini

Kit (QIAGEN, Hilden, Germany). The V3-V4 region of 16S rRNA genes was amplified using primers 338F and 806R (*Huse et al., 2007*) with TransStart Fastpfu DNA Polymerase (TransGen, Beijing, China) in 20 cycles. Three replicate PCR amplifications of each sample were purified with AxyPrep DNA Gel Extraction kit (AXYGEN, Union City, CA, USA), then pooled into equal concentrations after quantification. Next, $2 \times 300$ paired-end sequencing was performed for the equivalent pooled 16S rRNA PCR amplicons on an Illumina MiSeq instrument (San Diego, CA, USA).

## Bioinformatics and statistical analysis

Sequencing data was analyzed using Mothur (version 1.39.5) (*Schloss, Gevers & Westcott, 2011*) as previously described (*Zou et al., 2020*). In brief, the reads containing ambiguous bases, length shorter than 350 base pairs, with chimeric sequence or contaminant sequence were firstly removed. Then the SILVA reference database (*Quast et al., 2013*) (V132) was used as a reference for operational taxonomic units (OTUs) identification under the threshold of 97% similarity. Community richness, evenness, and diversity were assessed using Mothur. Differences between ASD and obesity samples were assessed by analysis of molecular variance (AMOVA). The taxonomic assignments were based on the Ribosomal Database Project (*Cole et al., 2009*) with the default parameter (80% threshold). Microbiota functions were predicted using phylogenetic investigation of communities by reconstruction of unobserved states (PICRUSt) (*Langille et al., 2013*). The significant differences in relative abundance of microbial taxa (OTU, genus, family, and phylum) and microbiota functional profiles between the ASD and obese groups were analyzed with STAMP using two-sided Welch's $t$-test (*Parks et al., 2014*). The coefficient relationship between species was calculated using R package with Spearman correlation algorithm, and the correlation parameters were set as: coefficient >0.35 or <-0.35 and $p < 0.05$ (*Taylor, 1990*).

## Accession numbers

The sequence data have been deposited in the National Omics Data Encyclopedia (NODE) under accession number OEX010410 (https://www.biosino.org/node/review/detail/OEV000113?code=KYM47EZL) and OEX010411 (https://www.biosino.org/node/review/detail/OEV000114?code=BS6WW5QC).

# RESULTS

## Bacterial composition in adult gut of ASD and obese subjects

A total of 42 fecal samples were collected from 21 adult patients with ASD and 21 obese adults. A total of 2,039,712 (39,341–59,610) high-quality 16S rRNA genes from 42 samples were contained by high-throughput DNA sequencing. To normalize the data and avoid statistical bias, 39,341 16S rRNA genes from each sample were chosen to calculate the richness, evenness, and diversity of bacterial community at 97% similarity. After the 42 samples were classified into two groups (ASD and Obesity), 12,411 OTUs were obtained (Supplemental File). The Good's coverage was over 99.8% in the two groups (Table 1), indicating that the sequencing depth was sufficient for studying the gut microbiota in adult individuals with ASD and obese adults.

**Table 1  Diversity evaluation of two groups microbiota.**

| Group | Sample | OTUs | Coverage | Richness | | Evenness | Diversity | |
|-------|--------|------|----------|----------|----------|----------|----------|----------|
| | | | | Chao | ACE | simpsoneven | Shannon | Simpson |
| ASD | 21 | 10511 | 0.999013 | 10837.95 | 10781.76 | 0.001651 | 5.223554 | 0.057639 |
| Normal weight ASD | 14 | 9444 | 0.997592 | 10027.12 | 10070.15 | 0.00335 | 5.485606 | 0.031612 |
| Obesity | 21 | 7530 | 0.998214 | 8544.02 | 8437.36 | 0.001693 | 4.338984 | 0.078443 |

## Microbiota of ASD and obesity

The total gut microbiota was examined by phylogenetic and taxonomic assessments of the 16S rRNA V3-V4 regions. Approximately 99.2% ($\pm$0.0041) of microbiota could be aligned to 18 phyla, 96.0% ($\pm$0.0364) to 98 families, and 87.1% ($\pm$0.0891) to 269 genera. At the phylum level, *Bacteroidetes* (average 48.5%, $\pm$0.221), *Firmicutes* (average 43.6%, $\pm$0.204), and *Proteobacteria* (average 2.93%, $\pm$0.051) were the three most abundant bacterial groups in the gut, which were common phyla in all samples (Table 2). At the family level, 15 families showed major abundance in two groups (>1% in at least one group, accounting for over 90% in each group, Table 3). Among the 15 families, *Lachnospiraceae*, *Prevotellaceae*, and *Bacteroidaceae* were dominant (>64% of each group). In the 269 identified genera, 47 genera were major genera (>0.1% in at least one group), including *Bacteroides*, *Prevotella*, *Megamonas*, *Roseburia*, *Lachnospiracea incertae sedis*, *Faecalibacterium*, and so on (Table 4). Among the major genera, seven ubiquitous (core) genera were consistently found across all analyzed samples and comprised an average of >23% of the total microbiota, including *Bacteroides*, *L. incertae sedis*, *Streptococcus*, *Ruminococcus2*, *Dorea*, *Blautia*, and *Clostridium XIVa*.

## Bacterial composition changes between ASD and obese groups

Among the 21 adult ASD, five were underweight (BMI<18.5) and two were obese (BMI>30). AMOVA analysis revealed that the gut microbiota composition among the three groups (obese ASD, underweight ASD and normal weight ASD) had no significant difference (Table 5, Fig. 1A), while the whole ASD group showed significant difference with obese group ($P_{\text{AMOVA}} < 0.05$). Hereinafter, we take all the 21 adult ASD as a whole to compare with obese group. Principal component analysis (Fig. 1B) showed that most subjects in the ASD and obese groups were distant from each group based on the gut microbiota composition. According to the evaluation of bacterial populations (Fig. 2, Table 1), subjects with ASD showed higher richness (ACE index and Chao index), higher evenness (Shannon even index), and higher diversity (Shannon and Simpson index). Thus, microbiota compositions differed between the ASD and obese groups, with the ASD showing higher biodiversity compared to the obese group.

At the phylum level, three major abundance phyla showed no significant variations between the ASD and obese groups. Only the phylum *Fusobacteria* was significantly decreased in ASD ($p = 0.031$) from 3.51% in the obese group to 0.10% in the ASD group. At the family level (Fig. 3), seven families showed significant differences between the ASD and obesity groups, six of which were major abundance families. The results showed that ASD was generally associated with the proportions of families. At the genus level (Table 4,
**Table 2  Significantly different phyla of gut microbiota between ASD and obesity.**

| phylum | ASD: mean rel. freq. (%) | ASD: std. dev. (%) | Obesity: mean rel. freq. (%) | Obesity: std. dev. (%) | *p*-values | Difference between means | 95.0% lower CI | 95.0% upper CI |
|---|---|---|---|---|---|---|---|---|
| *Firmicutes* | 49.74 | 18.31 | 37.55 | 20.09 | 5.18E−02 | 12.18 | −0.10 | 24.47 |
| *Actinobacteria* | 2.68 | 5.62 | 0.56 | 0.70 | 1.10E−01 | 2.11 | −0.52 | 4.75 |
| *Verrucomicrobia* | 1.01 | 4.47 | 0.01 | 0.04 | 3.28E−01 | 1.00 | −1.08 | 3.09 |
| *unclassified_Bacteria* | 0.94 | 0.45 | 0.67 | 0.29 | 3.34E−02 | 0.26 | 0.02 | 0.51 |
| *Synergistetes* | 0.08 | 0.34 | 0.00 | 0.00 | 3.19E−01 | 0.08 | −0.08 | 0.24 |
| *Tenericutes* | 0.00 | 0.00 | 0.00 | 0.00 | 2.14E−01 | 0.00 | 0.00 | 0.00 |
| *Planctomycetes* | 0.00 | 0.00 | 0.00 | 0.00 | 3.29E−01 | 0.00 | 0.00 | 0.00 |
| *Fibrobacteres* | 0.00 | 0.00 | 0.00 | 0.00 | 3.29E−01 | 0.00 | 0.00 | 0.00 |
| *Spirochaetes* | 0.00 | 0.00 | 0.00 | 0.00 | 3.29E−01 | 0.00 | 0.00 | 0.00 |
| *Ignavibacteriae* | 0.00 | 0.00 | 0.00 | 0.00 | 3.29E−01 | 0.00 | 0.00 | 0.00 |
| *Gemmatimonadetes* | 0.00 | 0.00 | 0.00 | 0.00 | 3.29E−01 | 0.00 | 0.00 | 0.00 |
| *Acidobacteria* | 0.00 | 0.00 | 0.00 | 0.00 | 1.84E−01 | 0.00 | 0.00 | 0.00 |
| *Chloroflexi* | 0.00 | 0.00 | 0.00 | 0.01 | 3.29E−01 | 0.00 | 0.00 | 0.00 |
| *Candidatus Saccharibacteria* | 0.00 | 0.00 | 0.01 | 0.01 | 6.98E−02 | 0.00 | −0.01 | 0.00 |
| *Lentisphaerae* | 0.00 | 0.01 | 0.06 | 0.27 | 3.47E−01 | −0.06 | −0.18 | 0.07 |
| *Elusimicrobia* | 0.00 | 0.00 | 0.20 | 0.88 | 3.29E−01 | −0.20 | −0.61 | 0.21 |
| *Proteobacteria* | 1.88 | 1.76 | 3.97 | 6.68 | 1.89E−01 | −2.09 | −5.29 | 1.10 |
| *Fusobacteria* | 0.10 | 0.24 | 3.52 | 6.57 | 3.08E−02 | −3.41 | −6.48 | −0.35 |
| *Bacteroidetes* | 43.56 | 20.92 | 53.44 | 21.67 | 1.50E−01 | −9.88 | −23.49 | 3.74 |

**Table 3  Major abundant and significantly different families in ASD and obesity gut microbiota.**

| family | feature | ASD | Obesity | Enriched in |
|---|---|---|---|---|
| *Bifidobacteriaceae* | Major & ubiquitous | 2.37% | 0.38% | |
| *Bacteroidaceae* | Major & ubiquitous | 10.87% | 19.43% | |
| *Porphyromonadaceae* | Major & difference | 1.01% | 0.47% | ASD |
| *Prevotellaceae* | Major & ubiquitous | 29.09% | 32.62% | |
| *Rikenellaceae* | Major | 1.96% | 0.11% | |
| *Streptococcaceae* | Major | 0.47% | 2.42% | |
| *Lachnospiraceae* | Major & difference | 25.89% | 12.42% | ASD |
| *Ruminococcaceae* | Major & difference | 11.91% | 5.68% | ASD |
| *Erysipelotrichaceae* | Major & difference | 2.71% | 0.52% | ASD |
| *Acidaminococcaceae* | Major | 1.47% | 0.83% | |
| *Veillonellaceae* | Major & difference | 2.09% | 13.95% | Obesity |
| *Fusobacteriaceae* | Major & difference | 0.09% | 3.51% | Obesity |
| *Sutterellaceae* | Major | 0.53% | 1.01% | |
| *Desulfovibrionaceae* | Difference | 0.35% | 0.03% | ASD |
| *Enterobacteriaceae* | Major & ubiquitous | 0.68% | 2.65% | |
| *Verrucomicrobiaceae* | Major | 1.01% | 0.01% | |

Fig. 4), 20 genera were found to significantly differ between ASD (16.62%) and obese

**Table 4  Major abundant and significantly different genera in difference gut microbiota.**

| Genus | Feature | ASD | Obesity | Enriched in |
|---|---|---|---|---|
| *Lachnospiracea_incertae_sedis* | Major & difference & ubiquitous | 4.28% | 1.70% | ASD |
| *Ruminococcus* | Major & difference | 2.81% | 0.37% | ASD |
| *Blautia* | Major & difference & ubiquitous | 3.28% | 1.08% | ASD |
| *Holdemanella* | Major & difference | 1.08% | 0.03% | ASD |
| *Clostridium IV* | Major & difference | 1.04% | 0.12% | ASD |
| *Ruminococcus2* | Major & difference & ubiquitous | 1.14% | 0.26% | ASD |
| *Clostridium XlVa* | Major & difference & ubiquitous | 1.44% | 0.65% | ASD |
| *Oscillibacter* | Major & difference | 0.26% | 0.07% | ASD |
| *Turicibacter* | Major & difference | 0.16% | 0.02% | ASD |
| *Bilophila* | Major & difference | 0.15% | 0.02% | ASD |
| *Odoribacter* | Difference | 0.05% | 0.01% | ASD |
| *Howardella* | Difference | 0.03% | 0.00% | ASD |
| *Senegalimassilia* | Difference | 0.03% | 0.00% | ASD |
| *Intestinibacter* | Difference | 0.02% | 0.00% | ASD |
| *Terrisporobacter* | Difference | 0.02% | 0.01% | ASD |
| *Intestinimonas* | Difference | 0.02% | 0.00% | ASD |
| *Holdemania* | Difference | 0.01% | 0.00% | ASD |
| *Murimonas* | Difference | 0.00% | 0.00% | ASD |
| *Fusobacterium* | Major & difference | 0.08% | 3.17% | Obesity |
| *Megamonas* | Major & difference | 0.70% | 11.77% | Obesity |
| *Bifidobacterium* | Major | 2.12% | 0.33% | |
| *Collinsella* | Major | 0.21% | 0.16% | |
| *Bacteroides* | Major & ubiquitous | 10.87% | 19.43% | |
| *Parabacteroides* | Major | 0.58% | 0.32% | |
| *Barnesiella* | Major | 0.15% | 0.05% | |
| *Prevotella* | Major | 27.82% | 30.21% | |
| *Paraprevotella* | Major | 0.16% | 0.06% | |
| *Alloprevotella* | Major | 0.94% | 2.24% | |
| *Alistipes* | Major | 1.95% | 0.11% | |
| *Elusimicrobium* | Major | 0.00% | 0.20% | |
| *Lactobacillus* | Major | 0.06% | 0.11% | |
| *Streptococcus* | Major & ubiquitous | 0.47% | 2.42% | |
| *Clostridium sensu stricto* | Major | 0.62% | 0.36% | |
| *Dorea* | Major & ubiquitous | 0.28% | 0.20% | |
| *Clostridium XlVb* | Major | 0.31% | 0.54% | |
| *Coprococcus* | Major | 1.23% | 0.42% | |
| *Roseburia* | Major | 5.40% | 2.95% | |
| *Anaerostipes* | Major | 0.12% | 0.19% | |
| *Fusicatenibacter* | Major | 0.65% | 0.34% | |
| *Butyrivibrio* | Major | 0.17% | 0.00% | |
| *Romboutsia* | Major | 0.49% | 0.22% | |
| *Faecalibacterium* | Major | 2.49% | 3.42% | |

**Table 4** (*continued*)

| Genus | Feature | ASD | Obesity | Enriched in |
|---|---|---|---|---|
| *Butyricicoccus* | Major | 0.19% | 0.25% | |
| *Gemmiger* | Major | 1.53% | 0.85% | |
| *Clostridium XVIII* | Major | 0.89% | 0.33% | |
| *Catenibacterium* | Major | 0.38% | 0.05% | |
| *Phascolarctobacterium* | Major | 1.47% | 0.80% | |
| *Dialister* | Major | 0.61% | 1.51% | |
| *Megasphaera* | Major | 0.55% | 0.14% | |
| *Mitsuokella* | Major | 0.13% | 0.27% | |
| *Parasutterella* | Major | 0.15% | 0.77% | |
| *Sutterella* | Major | 0.38% | 0.23% | |
| *Desulfovibrio* | Major | 0.16% | 0.01% | |
| *Escherichia/Shigella* | Major | 0.56% | 2.39% | |
| *Akkermansia* | Major | 1.01% | 0.01% | |

**Table 5  AMOVA analysis result between different groups based on microbiota composition.**

| Group1 | Group2 | *P* value |
|---|---|---|
| Normal weight ASD ($n = 14$) | Underweight ASD ($n = 5$) | 0.076 |
| Normal weight ASD ($n = 14$) | Obese ASD ($n = 2$) | 0.991 |
| Normal weight ASD ($n = 14$) | Obesity ($n = 21$) | 0.037[*] |
| Underweight ASD ($n = 5$) | Obese ASD ($n = 2$) | 0.674 |
| Underweight ASD ($n = 5$) | Obesity ($n = 21$) | 0.187 |
| Obese ASD ($n = 2$) | Obesity ($n = 21$) | 0.589 |
| ASD ($n = 21$) | Obesity ($n = 21$) | 0.032[*] |

**Notes.**
  *$P$ value $< 0.05$

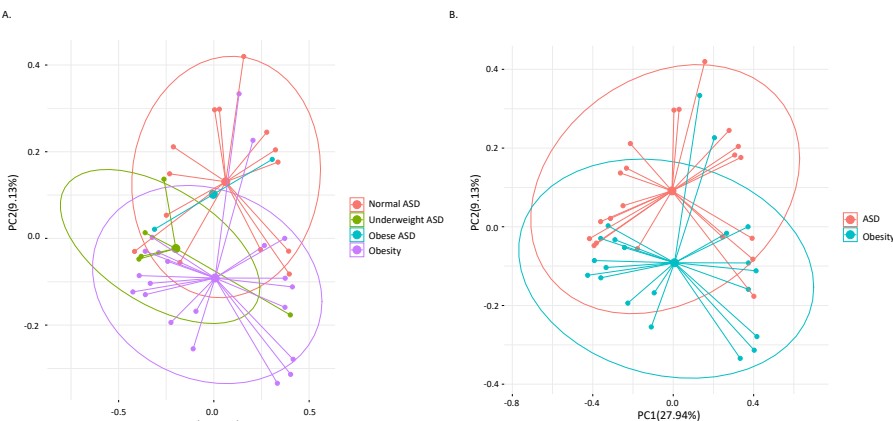

**Figure 1  Principal component analysis (PCA) calculated by weighted UniFrac distances.** (A) The 21 ASD adults were divided into three groups. (B) All the 21 ASD adults were taken as one group. Points representing samples were colored according to groups.

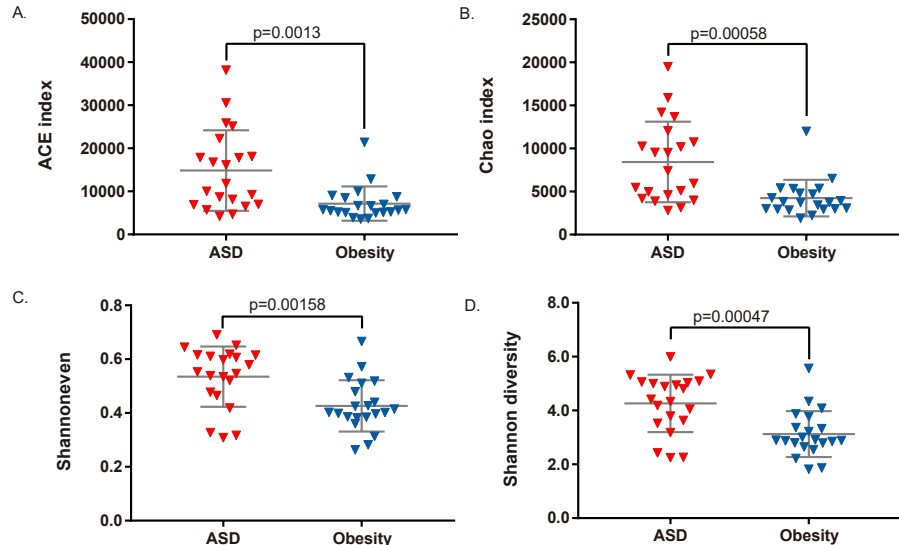

**Figure 2 Comparison of bacterial richness, evenness, and diversity between ASD and obesity groups.**
(A) ACE index, (B) Chao index, (C) Shannon evenness index, and (D) Shannon diversity index were compared by Student *t*-test.

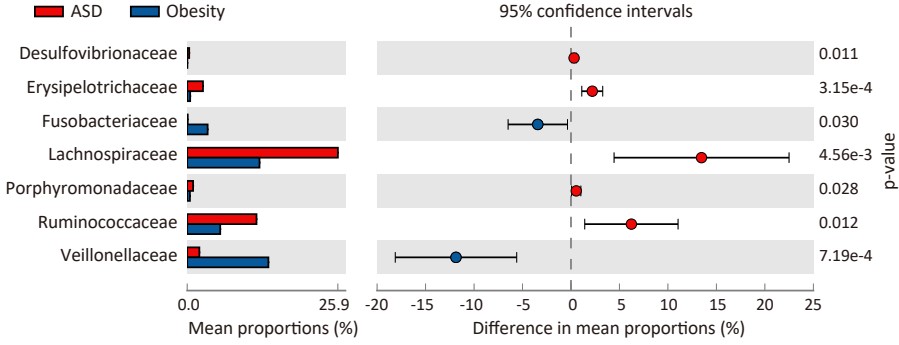

**Figure 3 Comparison of families between ASD and obesity microbiota.** The *p*-values was calculated based on two-sided Welch's *t*-test.

groups (19.28%), 12 of which were major genera. Only two genera were decreased in the ASD group: *Megamonas* and *Fusobacterium*.

At the species level (OTU from top 50, Table 6), nine abundant species significantly differed between ASD and obese subjects. Three were increased in the obesity gut microbiota, including *Megamonas funiformis*, *Fusobacterium mortiferum*, and *Dialister succinatiphilus*. Six species were increased in the ASD gut microbiota, including *Blautia wexlerae*, *Blautia faecis*, *Eubacterium eligens*, *Ruminococcus faecis*, *Phascolarctobacterium succinatutens*, and *Holdemanella biformis*.
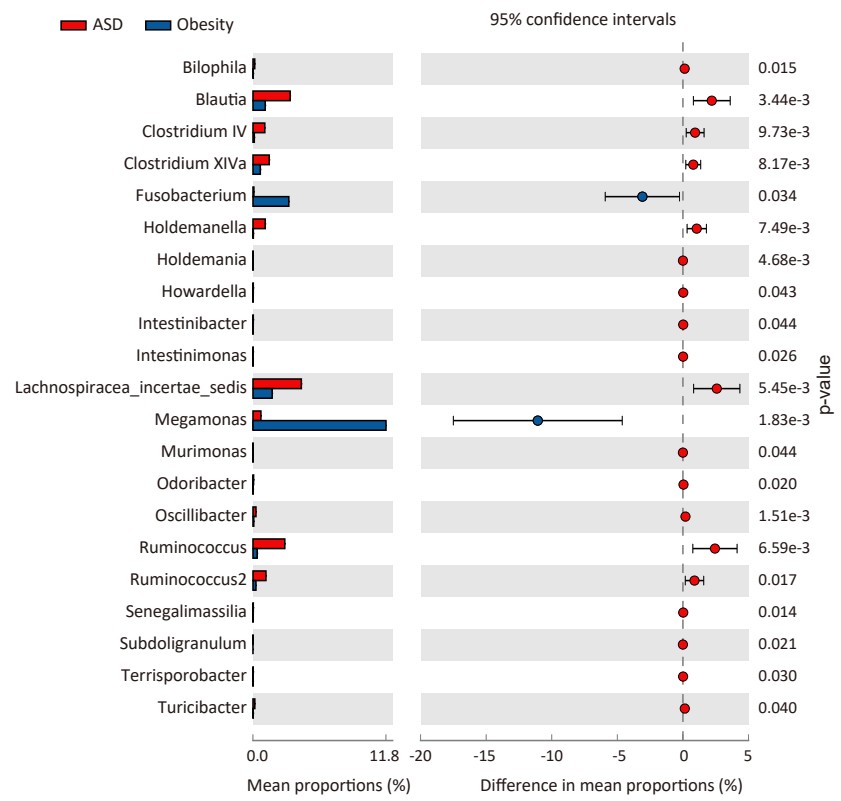

**Figure 4** **Comparison of genera between ASD and obesity microbiota.** The *p*-values was calculated based on two-sided Welch's *t*-test.

**Table 6** **Significantly different species of gut microbiota between ASD and obesity.**

| species | ASD | Control(Obesity) | p-values | Enriched in |
|---|---|---|---|---|
| *Blautia faecis* | 1.00% | 0.16% | 1.27E−03 | ASD |
| *Blautia wexlerae* | 1.18% | 0.33% | 1.14E−02 | ASD |
| *Dialister succinatiphilus* | 0.02% | 1.38% | 3.94E−03 | Obesity |
| *Eubacterium eligens* | 0.86% | 0.25% | 4.52E−02 | ASD |
| *Fusobacterium mortiferum* | 0.08% | 2.85% | 4.64E−02 | Obesity |
| *Holdemanella biformis* | 0.69% | 0.02% | 3.27E−02 | ASD |
| *Megamonas funiformis* | 0.51% | 10.35% | 2.25E−03 | Obesity |
| *Phascolarctobacterium succinatutens* | 0.68% | 0.08% | 6.61E−03 | ASD |
| *Ruminococcus faecis* | 0.69% | 0.12% | 1.88E−02 | ASD |
| *Prevotella copri* | 25.09% | 26.61% | 6.61E−01 | |

## Predicted functional potential change between ASD and obese microbiota

We used PICRUSt to predict the functional potential changes in ASD and Obesity (Table 7, Fig. 5). Thirty-three pathways differed between ASD and obese subjects, with 30 of which
**Table 7  Function prediction using PICRUSt based on 16S rRNA gene copy numbers.**

| Level 1 | Level 2 | pathway | p-value | |
|---------|---------|---------|---------|---|
| Environmental Information Processing | Membrane Transport | Bacterial secretion system | 1.70E−03 | C |
| Environmental Information Processing | Signaling Molecules and Interaction | Cellular antigens | 2.33E−02 | C |
| Environmental Information Processing | Signaling Molecules and Interaction | Ion channels | 9.92E−04 | C |
| Metabolism | Amino Acid Metabolism | Valine, leucine and isoleucine degradation | 3.70E−04 | C |
| Metabolism | Biosynthesis of Other Secondary Metabolites | Flavone and flavonol biosynthesis | 3.03E−02 | A |
| Metabolism | Biosynthesis of Other Secondary Metabolites | Isoquinoline alkaloid biosynthesis | 2.03E−03 | C |
| Metabolism | Biosynthesis of Other Secondary Metabolites | Penicillin and cephalosporin biosynthesis | 4.53E−02 | C |
| Metabolism | Enzyme Families | Protein kinases | 1.47E−02 | A |
| Metabolism | Glycan Biosynthesis and Metabolism | Glycosaminoglycan degradation | 4.03E−02 | C |
| Metabolism | Glycan Biosynthesis and Metabolism | Glycosphingolipid biosynthesis - ganglio series | 2.46E−02 | C |
| Metabolism | Glycan Biosynthesis and Metabolism | Lipopolysaccharide biosynthesis | 8.36E−04 | C |
| Metabolism | Glycan Biosynthesis and Metabolism | Lipopolysaccharide biosynthesis proteins | 4.95E−04 | C |
| Metabolism | Lipid Metabolism | Ether lipid metabolism | 8.60E−03 | A |
| Metabolism | Lipid Metabolism | Linoleic acid metabolism | 3.37E−02 | A |
| Metabolism | Lipid Metabolism | Primary bile acid biosynthesis | 6.39E−03 | A |
| Metabolism | Lipid Metabolism | Secondary bile acid biosynthesis | 5.97E−03 | A |
| Metabolism | Lipid Metabolism | Steroid hormone biosynthesis | 2.67E−02 | C |
| Metabolism | Metabolism of Cofactors and Vitamins | Riboflavin metabolism | 5.23E−03 | C |
| Metabolism | Metabolism of Cofactors and Vitamins | Ubiquinone and other terpenoid-quinone biosynthesis | 7.45E−03 | C |
| Metabolism | Metabolism of Other Amino Acids | D-Arginine and D-ornithine metabolism | 3.58E−03 | C |
| Metabolism | Metabolism of Other Amino Acids | Glutathione metabolism | 2.96E−03 | C |
| Metabolism | Metabolism of Other Amino Acids | Phosphonate and phosphinate metabolism | 1.41E−02 | A |
| Metabolism | Metabolism of Terpenoids and Polyketides | Geraniol degradation | 2.29E−03 | C |
| Metabolism | Metabolism of Terpenoids and Polyketides | Limonene and pinene degradation | 2.18E−02 | C |

belonging to metabolism and three pathways belonging to environmental information processing.

**Table 7** (*continued*)

| Level 1 | Level 2 | pathway | p-value | E |
|---|---|---|---|---|
| Metabolism | Metabolism of Terpenoids and Polyketides | Tetracycline biosynthesis | $1.39E{-}02$ | A |
| Metabolism | Xenobiotics Biodegradation and Metabolism | Atrazine degradation | $5.87E{-}03$ | A |
| Metabolism | Xenobiotics Biodegradation and Metabolism | Chloroalkane and chloroalkene degradation | $3.09E{-}03$ | A |
| Metabolism | Xenobiotics Biodegradation and Metabolism | Chlorocyclohexane and chlorobenzene degradation | $2.32E{-}02$ | A |
| Metabolism | Xenobiotics Biodegradation and Metabolism | Dioxin degradation | $1.04E{-}03$ | A |
| Metabolism | Xenobiotics Biodegradation and Metabolism | Ethylbenzene degradation | $6.98E{-}03$ | C |
| Metabolism | Xenobiotics Biodegradation and Metabolism | Styrene degradation | $4.36E{-}03$ | A |
| Metabolism | Xenobiotics Biodegradation and Metabolism | Toluene degradation | $3.14E{-}04$ | C |
| Metabolism | Xenobiotics Biodegradation and Metabolism | Xylene degradation | $5.69E{-}04$ | A |

## Correlations between bacterial species

To characterize the microbial interactions of ASD gut microbiota, correlation patterns of the top 10 species and different species between the ASD and obese groups were calculated (Table 8, Fig. 6, $p < 0.05$). In the ASD groups, 12 species showed correlations, including eight different species. In the obese group, 13 species showed correlations, including six different species. Four correlated species were shared by ASD and obese microbiota: *B. wexlerae*, *Blautia faecis*, *G. formicilis*, and *Bacteroides vulgatus*.

## Comparison of normal weight ASD and obesity

If we compare the alpha diversity, we can see that the normal weight ASD group ($n = 14$) had lower richness but higher diversity than the whole ASD group ($n = 21$) (Table 1), though this difference had no statistical significance. To exclude the affection of weight on gut microbiota, we then compare the normal weight ASD ($n = 14$) and obese group ($n = 21$).

At the phylum level, the increase of phylum *Firmicutes* and the decrease of phylum *Bacteroidetes* in the normal weight ASD showed statistical significance, which was not found when comparing all the ASD and obese group (Table 9). At the family level, the same seven families showed significant differences between the normal weight ASD and obesity groups (Table 9). At the genus level (Table 9), 16 genera were found to significantly differ with obese group, with genus *Allisonella* significantly decreased in the normal weight ASD group in addition *Megamonas* and *Fusobacterium*. At the species level, *Bacteroides plebeius* was significantly decreased in normal weight ASD, which was not observed when taking all the ASD as a whole (Table 9). Meanwhile, the abundance change of *Eubacterium eligens* and *Holdemanella biformis* showed no longer significance.

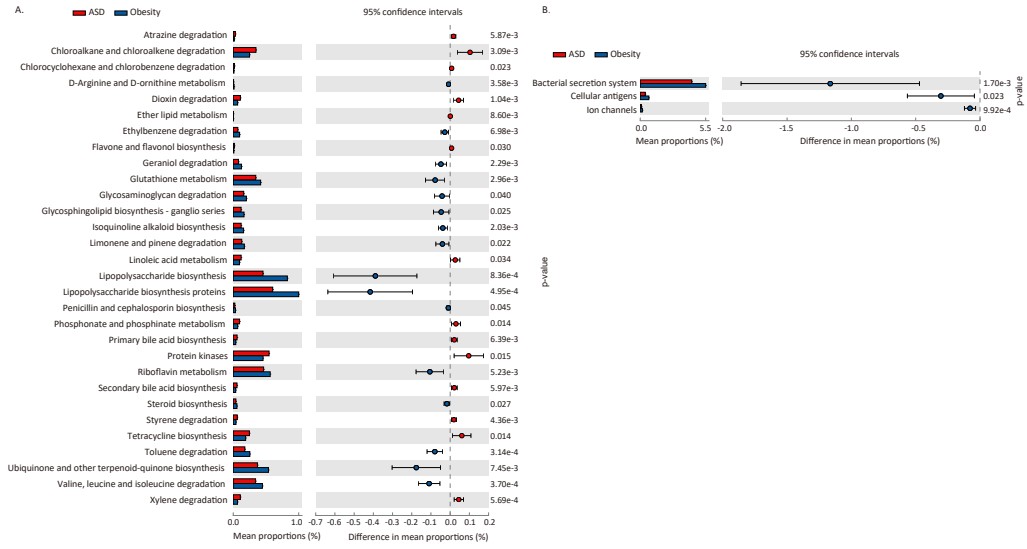

**Figure 5  Difference in functional pathway prediction using PICRUSt for ASD and obesity gut microbiota.** (A) Metabolism; (B) environmental information processing. The *p*-values was calculated based on two-sided Welch's *t*-test.

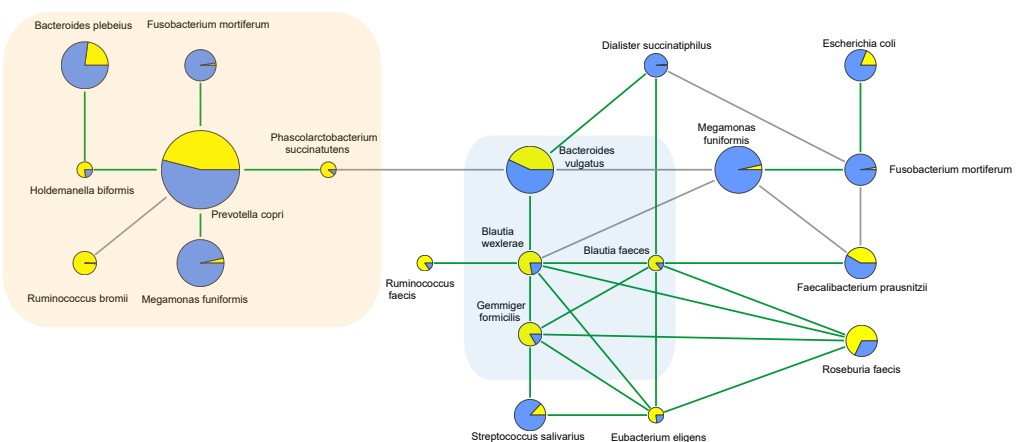

**Figure 6  Correlations between species calculated using Spearman correlation algorithm.** The light yellow part is the correlation species of ASD microbiota, and the light blue part is the shared correlation species by ASD and obesity microbiota, while the other part was the correlation species of obesity microbiota. The pie charts show relative species proportions in ASD (yellow) and obesity groups (blue), and the circle size represents the read number. Line color: Green (positive relationship) and grey (negative relationship).

## DISCUSSION

To characterize similarities and differences in the microbiota of adults with ASD and obese subjects, taxonomy assignments and difference analysis were performed between the two groups. In this study, we observed seven genera (*Bacteroides*, *Streptococcus*, *Dorea*, *L.*

**Table 8  Correlations of species calculated using Spearman algorithm.**

| | difference species | correlated species | spearmanCoef |
|---|---|---|---|
| ASD | Holdemanella biformis | Bacteroides plebeius | .537* |
| | Blautia wexlerae | Bacteroides vulgatus | .509* |
| | Dialister succinatiphilus | Bacteroides vulgatus | .479* |
| | Phascolarctobacterium succinatutens | Bacteroides vulgatus | -.632** |
| | Blautia wexlerae | Blautia faecis | .702** |
| | Dialister succinatiphilus | Blautia faecis | .459* |
| | Fusobacterium mortiferum | Prevotella copri | .550** |
| | Holdemanella biformis | Prevotella copri | .470* |
| | Megamonas funiformis | Prevotella copri | .446* |
| | Phascolarctobacterium succinatutens | Prevotella copri | .609** |
| Control(Obesity) | Megamonas funiformis | Bacteroides vulgatus | -.588** |
| | Blautia faecis | Roseburia faecis | .608** |
| | Blautia faecis | Faecalibacterium prausnitzii | .544* |
| | Blautia faecis | Eubacterium eligens | .481* |
| | Blautia wexlerae | Bacteroides vulgatus | .465* |
| | Blautia wexlerae | Roseburia faecis | .558** |
| | Blautia wexlerae | Blautia faecis | .464* |
| | Blautia wexlerae | Eubacterium eligens | .439* |
| | Blautia wexlerae | Ruminococcus faecis | .477* |
| | Eubacterium eligens | Roseburia faecis | .850** |
| | Eubacterium eligens | Streptococcus salivarius | .529* |
| | Fusobacterium mortiferum | Faecalibacterium prausnitzii | -.594** |
| | Fusobacterium mortiferum | Dialister succinatiphilus | -.511* |
| | Fusobacterium mortiferum | Escherichia coli | .504* |
| | Megamonas funiformis | Faecalibacterium prausnitzii | -.514* |
| | Megamonas funiformis | Blautia wexlerae | -.600** |
| | Megamonas funiformis | Fusobacterium mortiferum | .515* |

**Notes.**
*P < 0.05
**P < 0.01

incertae sedis, *Ruminococcus*, *Blautia*, and *Clostridium XIVa*) with an abundance of 23.75% (±0.0199 in two groups) as core microbiota (Table 4, Fig. 3). The microbiota of adults with ASD showed higher biodiversity than in obese control subjects; one phylum, seven families, 20 genera, and 13 species significantly differed between the two groups.

Previous studies suggested that *Bacteroidetes*, *Proteobacteria*, and *Fusobacteria* were enriched in children with ASD, while *Firmicutes* and *Actinobacteria* were lower in ASD (*Coretti et al., 2018*; *De Angelis et al., 2013*; *Ma et al., 2019*; *Zhang et al., 2018*). In the present study, though five phyla showed abundance changes of greater than 1%, only the relative abundance of phylum *Fusobacteria* showed a significant decrease in ASD group ($p < 0.05$) compared to the obese group (Table 2). Consistent with our findings, *Andoh et al. (2016)* reported a relative abundance of the phylum *Fusobacteria* increased in fecal of adults with obesity compared to lean people. The rising *Firmicutes/Bacteroidetes* (F/B) ratio has been suggested as an indicator of obesity, as *Koliada et al. (2017)* have reported

**Table 9  Significantly different taxa between normal weight ASD and Obesity in addition to all ASD and Obesity.**

| Taxonomy | p-values between normal ASD and Obesity | Abundance (%) | | | p-values between all ASD and Obesity | |
|---|---|---|---|---|---|---|
| | | normal ASD ($n=14$) | Obesity ($n=21$) | all ASD ($n=21$) | | |
| Firmicutes | 0.0118 | 54.23 | 37.55 | 49.74 | 0.0518 | phylum |
| Bacteroidetes | 0.0255 | 37.26 | 53.44 | 43.56 | 0.1503 | phylum |
| Fusobacteria | 0.0324 | 0.13 | 3.52 | 0.10 | 0.0308 | phylum |
| Desulfovibrionaceae | 0.0333 | 0.37 | 0.03 | 0.35 | 0.0105 | family |
| Erysipelotrichaceae | 0.0007 | 3.4 | 0.52 | 2.71 | 0.0003 | family |
| Fusobacteriaceae | 0.0317 | 0.12 | 3.51 | 0.09 | 0.0304 | family |
| Lachnospiraceae | 0.0038 | 29.29 | 12.42 | 25.89 | 0.0046 | family |
| Porphyromonadaceae | 0.0113 | 1.28 | 0.47 | 1.01 | 0.028 | family |
| Ruminococcaceae | 0.0218 | 12.33 | 5.68 | 11.91 | 0.0125 | family |
| Veillonellaceae | 0.0004 | 1.37 | 13.95 | 2.09 | 0.0007 | family |
| Bilophila | 0.024 | 0.19 | 0.02 | 0.15 | 0.0146 | genus |
| Blautia | 0.0097 | 3.65 | 1.08 | 3.283 | 0.0034 | genus |
| Clostridium IV | 0.0209 | 1.38 | 0.1 | 1.04 | 0.0097 | genus |
| Clostridium XlVa | 0.0137 | 1.74 | 0.65 | 1.44 | 0.0082 | genus |
| Fusobacterium | 0.0355 | 0.1 | 3.17 | 0.08 | 0.0343 | genus |
| Holdemanella | 0.0225 | 1.39 | 0.03 | 1.08 | 0.0075 | genus |
| Holdemania | 0.0227 | 0.005 | 0.001 | 0.005 | 0.0047 | genus |
| Howardella | 0.1026 | 0.028 | 0.002 | 0.03 | 0.0431 | genus |
| Intestinibacter | 0.0573 | 0.03 | 0.002 | 0.02 | 0.0444 | genus |
| Intestinimonas | 0.086 | 0.01 | 0.002 | 0.02 | 0.0263 | genus |
| Lachnospiracea_incertae_sedis | 0.0025 | 5.19 | 1.69 | 4.28 | 0.0054 | genus |
| Megamonas | 0.0016 | 0.52 | 11.77 | 0.7 | 0.0018 | genus |
| Murimonas | 0.1054 | 0.0007 | 0.0001 | 0.0008 | 0.0437 | genus |
| Odoribacter | 0.0333 | 0.07 | 0.01 | 0.05 | 0.0197 | genus |
| Oscillibacter | 0.002 | 0.3 | 0.07 | 0.26 | 0.0015 | genus |
| Ruminococcus | 0.0663 | 2.76 | 0.37 | 2.81 | 0.0066 | genus |
| Ruminococcus2 | 0.0292 | 1.46 | 0.26 | 1.14 | 0.0165 | genus |
| Senegalimassilia | 0.0344 | 0.03 | 0.002 | 0.03 | 0.0144 | genus |
| Terrisporobacter | 0.0282 | 0.02 | 0.006 | 0.02 | 0.0303 | genus |
| Turicibacter | 0.0457 | 0.21 | 0.02 | 0.16 | 0.0402 | genus |
| Allisonella | 0.0298 | 0.004 | 0.04 | 0.009 | 0.0552 | genus |
| Bacteroides plebeius | 0.0233 | 0.17 | 5.59 | 1.68 | 0.1571 | species |
| Blautia faecis | 0.0023 | 0.94 | 0.16 | 1 | 0.0013 | species |
| Blautia wexlerae | 0.0229 | 1.42 | 0.33 | 1.18 | 0.0114 | species |
| Dialister succinatiphilus | 0.0037 | 0.01 | 1.38 | 0.02 | 0.0039 | species |
| Eubacterium eligens | 0.0608 | 0.95 | 0.25 | 0.86 | 0.0452 | species |
| Fusobacterium mortiferum | 0.0479 | 0.09 | 2.85 | 0.08 | 0.0464 | species |
| Holdemanella biformis | 0.0629 | 0.88 | 0.02 | 0.69 | 0.0327 | species |
| Megamonas funiformis | 0.002 | 0.37 | 10.35 | 0.51 | 0.0022 | species |

**Table 9** (*continued*)

| Taxonomy | p-values between normal ASD and Obesity | Abundance (%) | | | p-values between all ASD and Obesity | |
|---|---|---|---|---|---|---|
| | | normal ASD (*n* = 14) | Obesity (*n* = 21) | all ASD (*n* = 21) | | |
| *Phascolarctobacterium succinatutens* | 0.0253 | 0.57 | 0.08 | 0.68 | 0.0066 | species |
| *Ruminococcus faecis* | 0.0399 | 0.86 | 0.12 | 0.69 | 0.0188 | species |

that a higher abundance of *Firmicutes* and a lower level of *Bacteroidetes* in adults with obesity than in normal-weight adults in Ukraine. But in our study, the proportion of F/B was significantly higher in adults with ASD (1.14) than that in adults with obesity (0.70) ($p < 0.05$, Wilcoxon rank-sum test) (Table 2). Although dietary habits have been proposed to contribute to this ratio difference (*Zhang et al., 2018*), age may be also involved. Consequently, we conjectured that the proportion of F/B may be closely associated with both ASD and obesity.

At the family level, we observed *Lachnospiraceae, Ruminococcaceae, Erysipelotrichaceae, Porphyromonadaceae*, and *Desulfovibrionaceae* were enriched in adults with ASD, while *Fusobacteriaceae* and *Veillonellaceae* were significantly decreased; *Prevotellaceae* was dominant family both ASD (28.9%) and obese (32.5%) groups (Table 3). *Serena et al. (2018)* have indicated that the families *Veillonellaceae* and *Prevotellaceae* were significantly increased in obese individuals compared to healthy subjects, which are major bacteria succinate-producing (*Nakayama et al., 2017*). In adipose tissue, succinate possesses antilipolytic actions through binding to cognate receptor succinate receptor 1 (Sncr1), and leads to fat accumulation (*McCreath et al., 2015*). Therefore, *Veillonellaceae* and *Prevotellaceae* were playing an important role in development of obesity. Compared with non-obese adults with ASD, we found that a higher abundance of *Pseudomonaceae, Prevotellaceae*, and *Fusobacteriaceae*, as well as a lower abundance of *Lachnospiraceaea* and *Ruminococcaceae* in fecal of obese adults. These results were consistent with previous studies on gut microflora in appendix samples of obese patients (*Moreno-Indias et al., 2016*). As is known, the families *Lachnospiraceaea* and *Ruminococcaceae* were able to ferment carbohydrates to produce short-chain fatty acids (SCFAs) which mainly includes acetic acid, propionic acid and butyric acid (*Biddle et al., 2013*). Among SCFAs, butyrate can inhibit the release of pro-inflammatory cytokines like TNF-$\alpha$ and IL-6 and play an anti-inflammatory role (*Lewis et al., 2010*).

At the genus level, only two genera (*Megamonas* and *Fusobacterium*) were significantly decreased, while 18 genera were increased in the ASD group. The abundance of *Megamonas* decreased from 11.67% in obese group to only 0.7% in adults with ASD (Table 4, Fig. 4). It had been reported that *Megamonas* can ferment glucose into acetic and propionic acid, which has been shown to be a substrate for lipogenesis and cholesterol formation and serve as an energy source for the host (*Kieler et al., 2017*). Consistent with our findings, previous study reported that *Megamonas* was enriched in obese adults, which was positively associated with obesity (*Chiu et al., 2014*; *Maya-Lucas et al., 2019*). Additionally, *Andoh et al. (2016)* have reported that the genus *Fusobacterium* was significantly enriched in obese individuals compared to lean people. The genus *Fusobacterium* belongs to the phylum

*Fusobacteria*, which may be involved into the occurrence and development of obesity by inducing the host's inflammatory response (*Kostic et al., 2012*). Furthermore, some studies have indicated that the genus *Fusobacterium* was closely associated with obesity-related colorectal neoplasms (*Amitay et al., 2017*; *McCoy et al., 2013*), which may be one of the mechanisms that obese people are prone to tumors. Thus, we inferred that these genera are strongly associated with obesity and may be potential pathogens.

Consistent with previous studies of children with ASD (*Berding & Donovan, 2018*), we found that adults with ASD had a higher abundance of *Ruminococcus* (2.44% increase). Previous study has shown that the genera *Ruminococcus* could produce butyrate and alleviates insulin resistance, which was beneficial to control obesity (*Gao et al., 2018*).

*Senegalimassilia*, belonging to family *Coriobacteriaceae*, together with *Clostridium XIVa*, has been identified as a p-cresol-producing intestinal bacteria (*Saito et al., 2018*). p-Cresol can inhibit dopamine beta-hydroxylase (*Southan, De Wolf Jr & Kruse, 1990*), an enzyme catalyzing the hydroxylation of dopamine to norepinephrine, which functions as a neurotransmitter. p-Cresol may modulate behavioral abnormalities and autism severity, and high levels of p-cresol are often observed in children with ASD (*Persico & Napolioni, 2013*).

As a major genus in both groups, *Blautia* was significantly decreased (2.2% decrease) in obese group. *Blautia* plays an important role in nutrient assimilation, gut maturation, and mucosal serotonin synthesis in the gut which accelerates gastrointestinal motility (*Golubeva et al., 2017*; *Liu et al., 2019*). Agreement with our results, a previous study has indicated the genus *Blautia* was decreased in obese adults, which was inversely association with visceral fat accumulation (*Ozato et al., 2019*). So it may be a potential a potentially beneficial genus for obese patients.

In our study, we also observed that the major genera *Butyricicoccus*, *Clostridium IV*, *Parasutterella*, *Parabacteroides*, and *Roseburia* were decreased in obese group (Table 4). Interestingly, recent study has shown that these genera were negatively associated with host's BMI and lipid levels (*Zeng et al., 2019*). Among these genera, *Butyricicoccus* (*Takada et al., 2016*), *Clostridium IV* (*Moens & De Vuyst, 2017*) and *Roseburia* (*Kasahara et al., 2018*) can produce butyrate which has anti-inflammatory functions, thus being beneficial to anti-obesity. Moreover, an animal study has shown that *Parabacteroides* is beneficial for reducing host weight and hyperglycemia (*Wang et al., 2019*). Therefore, above-mentioned bacteria may be beneficial in controlling obesity.

At the species level, two *Blautia* species (*B. wexlerae* and *B. faeces*) and *R. faecis* were significantly increased in adults with ASD (Table 6). *Kasai* et al. observed that *B. wexlerae* was significantly reduced in *obese* group compared to non-obese (*Kasai et al., 2015*). *B. wexlerae* is also a major acetate producer (*Jang et al., 2019*). When the abundance of *B. wexlerae* decreased, the production of acetate and butyric acid was also decreased (*Jang et al., 2019*; *Vital et al., 2018*). However, animal experiment has shown that butyrate can improve insulin resistance and reduce fat accumulation (*Khan & Jena, 2016*). Therefore, this may be one of the mechanisms of *B. wexlerae* anti-obesity.

Additionally, we also observed that the *Dialister succinatiphilus*, *Megamonas funiformis* and *Fusobacterium mortiferum* were enriched in obese group. *M. funiformis* and *F.*

mortiferum belong to Gram-negative bacteria (*Sakon et al., 2008*), which their cell walls contain more lipopolysaccharides that can induce or aggravate the host to produce inflammatory response and insulin resistance, thus involving in the occurrence and development of obesity (*Muscogiuri et al., 2019*). Additionally, study indicated that *D. succinatiphilus* is a succinate-utilizing bacteria (*Nakayama et al., 2017*). Morotomi et al. have shown that succinate can stimulate the growth and reproduction of *D. succinatiphilus*, while producing a large amount of propionate (*Morotomi et al., 2008*). Interestingly, Ren et al. have found that circulating succinate concentrations was increased in patients with obesity or type 2 diabetes (*Ren et al., 2019*). Moreover, Ceperuelo-Mallafré et al. have indicated that succinate concentrations was significantly decreased in serum of patients with diabetes after bariatric surgery, and considered baseline succinate levels to have an independent predictive effect on diabetic remission (*Ceperuelo-Mallafre et al., 2019*). Therefore, we speculated that the relative abundance of *D. succinatiphilus* may be as a biomarker for predicting obesity.

Consistent with our findings, *Serena et al. (2018)* observed a lower abundance of *Phascolarctobacterium spp*. in obese individuals than in non-obese people, which is known as succinate-utilizing bacterium that may be affecting the energy metabolism of the host by participating in the metabolism of succinate, thus reducing the occurrence of obesity in the host. Interestingly, *Phascolarctobacterium succinatuten*, an asaccharolytic bacteria distributed broadly in the gastrointestinal tract, can utilize succinate generated by other intestinal bacterial species to produce propionate (*Watanabe, Nagai & Morotomi, 2012*), which can cross the blood–brain barrier and act as a neurotoxin to elicit ASD-like behavior (*Berding & Donovan, 2016*).

In addition, the most abundant species in both groups was *Prevotella copri*, which showed no significant difference between the ASD and obese groups (Table 6). Some studies have shown that *P. copri* were involved in occurrence of obesity (*Stanislawski et al., 2019*) through promoting the biosynthesis of branched-chain amino acids to induce insulin resistance (*Pedersen et al., 2016*) and stimulating the secretion of inflammatory factors to trigger or aggravate the host's inflammatory response (*Larsen, 2017*). Therefore, *P. copri* is associated with both ASD and obesity, which may be a common-owned biomarker of ASD and obesity.

This is the first study to compare the microbial composition between ASD patients and obesity adults. Nevertheless, the current study has several limitations. First, the sample size of the study was relatively small, and both underweight ($n = 5$) and obese ($n = 2$) adult ASD were included in the 21 patients. The small sample size limited our further grouping and comparison between obese ASD and obesity. Indeed, we performed a comparison analysis between normal weight ASDs ($n = 14$) and obesity, and most of the significant changes was similar as observed in the comparison between all ASDs ($n = 21$) and obesity. However, the conclusion of this study is somewhat weakened, and more ASD adults including obese ASD will be recruited in subsequent studies. Second, all the DNA were extracted using QIAamp DNA stool mini kit, which had no bead-beating step and was hard to lyse Gram-positive bacteria (*Albertsen et al., 2015*; *Guo & Zhang, 2013*). This might cause gram-positive bacteria to be underrepresented. Though it is not critical to the conclusions

since all the samples were processed similarly, a kit with bead-beating step is preferred. Third, the diets of obese group were not uniform. Since the type of diet has a great influence on the gut microbiota, the dietary data should be collected and analyzed in further studies.

## CONCLUSIONS

In the present study, 42 fecal samples were collected from 21 adult patients with ASD and 21 obese adults. The gut microbiota composition was analyzed and compared to existing reports of children with ASD or obesity. We found that the microbiota in adults with ASD exhibited higher biodiversity than that of obese controls, with one phylum, seven families, 20 genera, and nine species showing significant differences between the two groups. The two genera (*Megamonas* and *Fusobacterium*) were significantly enriched in obese group. The propionate-producing species *P. succinatuten* increased in adults with ASD. The species *D. succinatiphilus* may be as a biomarker for predicting obesity, as well as *P. copri* may be a common-owned biomarker of ASD and obesity. Furthermore, we observed that the unique intestinal microbiota is strongly related to the occurrence and development of ASD or obesity, making the microbiota a potential treatment target for patients with ASD or obese patients. More importantly, compared to previous reports, we observed some conflicting results because of the different ages and obesity status of the patients with ASD, which should be examined in further studies.

## ACKNOWLEDGEMENTS

We would like to acknowledge all the participants and their families who kindly took part in this research.

### Funding

This work was supported by the Development Fund for Shanghai Talents [Grant number 201567]. The funders had no role in study design, data collection and analysis, decision to publish, or preparation of the manuscript.

### Grant Disclosures

The following grant information was disclosed by the authors:
Development Fund for Shanghai Talents: 201567.

### Competing Interests

The authors declare there are no competing interests.

### Author Contributions

- Qiang Zhang performed the experiments, analyzed the data, prepared figures and/or tables, authored or reviewed drafts of the paper, and approved the final draft.
- Rong Zou performed the experiments, analyzed the data, prepared figures and/or tables, and approved the final draft.

- Min Guo and Mengmeng Duan performed the experiments, prepared figures and/or tables, and approved the final draft.
- Quan Li conceived and designed the experiments, authored or reviewed drafts of the paper, and approved the final draft.
- Huajun Zheng conceived and designed the experiments, analyzed the data, prepared figures and/or tables, authored or reviewed drafts of the paper, and approved the final draft.

### Human Ethics

The following information was supplied relating to ethical approvals (i.e., approving body and any reference numbers):

The Medical Ethical Committee of Shanghai Institute of Planned Parenthood Research approved the study (NO: PJ2019-17).

### Data Availability

The sequence data are available in the National Omics Data Encyclopedia (NODE): OEX010410, OEX010411.

https://www.biosino.org/node/review/detail/OEV000113?code=KYM47EZL.
https://www.biosino.org/node/review/detail/OEV000114?code=BS6WW5QC.

### Supplemental Information

Supplemental information for this article can be found online at http://dx.doi.org/10.7717/peerj.10946#supplemental-information.

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
