# Peer review of "Comparison of gut microbiota between adults with autism spectrum disorder and obese adults"

_PeerJ, doi:10.7717/peerj.10946_

## Round 0.1 · original submission · Major Revisions

Please do study in detail the comments and revise accordingly as the fundamentals are weak.

·

Basic reporting

Good.

Experimental design

1. Please describe the inclusion criteria and exclusion criteria in detail;
2. Please specify the time and frequency of fecal collection. Please note that the time taken and dietary conditions could affect the structure of feces;
3. Please pay special attention to the effect of antibiotics before enrolled in the study;
4. Please indicate whether the enrolled autistic patients have gastrointestinal symptoms;

Validity of the findings

this study is a small sample single center study, so please give the conclusion carefully.

Additional comments

This study compared the difference of intestinal flora between autistic and obese patients, which is of great significance for clinical improvement of comprehensive treatment of autism and obesity. However, in order to obtain clearer information, the authors are requested to supplement the following information:
1. Please describe the inclusion criteria and exclusion criteria in detail;
2. Please specify the time and frequency of fecal collection. Please note that the time taken and dietary conditions could affect the structure of feces;
3. Please pay special attention to the effect of antibiotics before enrolled in the study;
4. Please indicate whether the enrolled autistic patients have gastrointestinal symptoms;
In addition, this study is a small sample single center study, so please give the conclusion carefully.

Reviewer 2 ·

Basic reporting

Overall, the text reads rather well but English language still need significant revisions in terms of grammar and spelling mistakes. Introduction may want to include the changes to autism in adults, as they transit from childhood including metabolic co-morbidity like obesity. Any studies reporting gut microbiota changes in autistic obese child and obese adults – any differences? How does obesity affect autism in adults? In general, the structure of the paper conforms to the PeerJ requirements. Likewise, figures are relevant and of sufficient qualities, although larger texts for figure 6 are preferable. Raw data is provided as per journal policy.

Experimental design

The study recruited ASD with normal BMI but I note some were obese (max BMI was 31.9). While I like the general idea of this study but I am not entirely clear on the intention of experimental design. Does the researcher hope to differentiate obese ASD from obesity, if this is the case, then researcher should recruit obese ASD. If the intention is to differentiate ASD from obesity, then there shouldn’t be any obese ASD adults or to have different groups of ASD i.e. non-obese ASD and obese ASD with obesity as controls. Again, the grouping depends on the research hypothesis which I am not clear of. The psychological status of ASD adults has not been provided for e.g. the autism spectrum quotient. Likewise the medical history and metabolic status of ASD and obese adults are not provided.

Validity of the findings

The study findings do provide a couple of interesting observations. Limitations of the study are not provided in the discussion. What is the significance of higher biodiversity in ASD vs. obese? Due to grouping design as described above, the impact of results somehow become limited since the findings become somewhat expected, e.g., fusobacteria is enriched in obesity and contradictory, e.g., enriched fusobacteria in children but reduced in adults.

Additional comments

While the idea is good, but study design somehow limits the usefulness of findings.

·

Basic reporting

No comment

Experimental design

I have a few comments on the methodology, which can be seen in the general comments for the author section.

Validity of the findings

No comment.

Additional comments

This is a well-made study investigating the gut microbiota of adults with ASD, as compared to obese adults. The authors rightfully conclude that certain bacteria differs between these two groups, and thus future experiments on gut microbiota in ASD need to take the BMI or growth curves into account. The article is well written, and thus the information are easily accessible.
However, I do have a couple of comments, mainly on the methodology. These issues does not affect the conclusion of the article, but need to be mentioned, possibly in a limitation section.

L. 61-62: Since the manuscript focuses on adults, I will recommend deleting the definition of obesity in children.

L. 80. “… which can lead to the ASD-like behaviors, language development deficits, and so on”. This comment needs a reference.

L. 97: “…comparison of intestinal microbiota characteristics between patients with ASD and obese patients are rare”. Does any study exists investigating this? If not, the authors should write that they are not aware of studies investigating this or something like that. “rare” suggest that other such studies exist.

L. 104. I have a couple of questions regarding the study population with ASD.
- How was the ASD diagnosed in these patients? K-SADS? ADOS?
- The authors have included 21 patients diagnosed with ASD and 21 obese persons. It would have been beneficial to have included a third group, that was obese and had ASD, when trying to discuss the effects of obesity versus ASD. This is not critical to the study, but would be beneficial for future studies.
- BMI of ASD cases. You have a BMI range going from underweight (15.9) to obese (31.9) amongst your patients with ASD. Could these extremes have different gut microbiota? E.g. did the obese patient(s?) with ASD have a gut microbiota more like the obese case group?

L. 109: The authors included obese adults from a gym. Are these obese adults in the process of losing weight, since they were recruited at the gym? In that case, were they currently undergoing a diet? The exercise as well as introduction of low calory diets, may have impacted their gut microbiota.

L. 110: I would like more information on how the samples were collected? At the research lab, or in the home of study participants? Were samples immediately (how quick?) transferred to -80 °C, or were they transported to the lab on cold ice?

L. 117. The authors mention that they used the QIAamp DNA stool mini kit. Gram-positive bacteria are notoriously difficult to extract DNA from, and often require additional steps to process (see pmid: 26182345, as well as the new preprint on bioRxiv: https://www.biorxiv.org/content/10.1101/2020.06.15.151753v1 for details). Often this include bead-beating while others utilize extended chemical lysis steps. This is not critical to the conclusions of this study, since all samples were processed similarly, but it might have caused gram-positive bacteria to be underrepresented. This should be mentioned in a limitation section.
If the authors prefers QIAGEN as provider, they have released a PowerFecal kit. This is a good alternative to QIAamp that can be automated on the same machinery, but which does include bead-beating.

Figure 4+5+6 – what test do the authors use to compare the differences in mean abundance between ASD and obesity?

L225-227. “At the phylum-> Figure 1).” Since the authors does not use these information in the subsequent discussion, and has already mentioned it in the results, I think these three lines can be deleted from the discussion.

L. 258. “As all know…”. May I suggest using another phrase?

L. 266-268. That the reason Megamonas was more abundant amongst obese patients, was due to its ability to produce substrates for lipogenesis, is interesting, and probably not wrong. However, it is stated as a fact in the section. It should be more clear, that it is a speculation by the authors.

L. 281-282: Yes, leaky gut has been suggested to be involved in ASD. But since leaky gut has not been mentioned previously in the manuscript, and is not mentioned later, the speculated relationship between increased Ruminococcus and leaky gut, does not really fit into the rest of the article.

L. 346-351: The authors mentioned that they observed enrichment with P. copri in both obese patients and patients with ASD. Conversely Kang et al. reported a reduced relative abundance of this bacteria. The authors uses this to conclude that children and adults with ASD may have different microbiota compositions. However, gut microbiota can be altered in a lot of ways, meaning that the differences in gut microbiota between this study and the study by Kang et al., may be a result of differences in study population, lifestyle, methodology or just normal variations in gut microbiota. Unless the authors compares the gut microbiota in children and adults with ASD in parallel, I would hesitate to make that conclusion with confidence. Furthermore, the ASD of children are not the focus on this study, and thus I am unsure why the authors wish to comment on differences between adults and children.

While the study seems well designed and the article is well written, it would benefit from a limitation section. This should include, but not necessarily be limited to:
- The wide range in BMI amongst the included patients with ASD.
- The risk of behavioral/dietary modifications that may arise from recruiting patients at a gym.
- The potential underrepresentation of gram-positive bacteria due to lack of mechanical disruption.

A couple of minor typos
- L. 91. “..are positive correlated with obese..” – this should be replaced with “are positively correlated with obesity…”
- L. 95: “…studies have shown that…” replace “have shown” with either reporting/documenting/showing.
- L. 218. “common” – does the authors means “similarities”?
- L. 334. “reched”. I am unsure what the authors mean with this word. It is probably just a typo.
- L. 366. “specie” – replaced with “species”. While “specie” is correct for singular form in the original Latin version, the meanings have diverged and currently, “species” is usually as both singular and plural.

---

## Round 0.2 · Major Revisions

Please edit this manuscript as per the second peer reviewer's comments.

Reviewer 2 ·

Basic reporting

The authors have revised accordingly to my comments.

Experimental design

The authors have revised accordingly and provided limitations as suggested.

Validity of the findings

The authors have revised accordingly and provided additional analysis, which is acceptable in its current form.

Additional comments

The current version is much improved and is suitable for publication in its current form.

·

Basic reporting

I observed a potential error with the reference "Association, 2013"
I can see that the authors utilize reference management software – which is highly recommended. However, it seems that an error has occurred when entering the data for the American Psychiatric Association. The software has understood it as a name, and thus only used the last part of the name. I do not know what software the authors use, although I know that it is a known problem in Mendeley – here, it can be solved when entering the name, by clicking the “institution” option in the dropdown menu below. I am not sufficiently familiar with other reference management softwares to guide in these.

There are some minor typos and and grammatical errors, listed below:
L. 65: “…were obesity”. This should be “…were obese”.

L. 69: “Recent decades”. This should be “In recent decades”.

L. 75: “though regulating”. This should be “through regulation”.

L. 81. “Additionally, other study”. This should be “Additionally, another study”

L. 82 + l. 100: Replace “mirobiota” with “microbiota”.

L. 91: Replace “Though” with “Through”

L. 109 “With a mean ages years (ranging…” Something is wrong with this sentence. It seems like the () has been misplaced.

L. 115: “in one month prior” this should be “for one month prior”.

L. 117-119: “Twenty-one obese adults (…) …neurophychiatric diseases,”. Rephrase this sentence. This could be like this: “Twenty-one gender and age matched obese adults (…) that did not suffer from ASD, other neurodevelopmental disorders or neuropsychiatric diseases”.

L. 243-245: “only the phylum Fusobacteria… enriched in obese group”. When you see a decrease amongst ASD, it makes sense that you see an enrichment in the obese groups, since you are comparing those two groups. I would recommend that you just write that the relative abundance of phylum Fusobacteria were lower amongst your ASD group compared to the obese group.

L. 248-251: “Furthermore, Koliada… BMI(Kolliada et al., 2017).” This sentence is a bit unclear.

Experimental design

L. 192-196. I appreciate the information on the weight distribution of the ASD group. However, I do not think, that no significant difference in AMOVA are a sufficiently strong argument to group all ASDs. Despite no overall difference in microbiota composition, you might still see differences among the less dominant bacteria. Especially considering the fact that the individual subgroups was small when performing the AMOVA. I would recommend trying to perform the analyses again (alpha diversity, PCoA, AMOVA, and potentially the Picrust), but only for the group consisting of normal weight ASDs. Then see if the results are still the same as what you observed for all ASDs combined. The results of this subanalysis can then be either placed in the supplementary, or, if the results were indeed similar, just mention that you perform this subanalysis, and that it provided no differences.

Validity of the findings

No Comment

Additional comments

L. 247: Kostic et al, 2012 only reports on the species Fusobacterium nucleatum, that belongs to the Fusobacteria phylum. I would not suggest extrapolating characteristics from a single species to the whole phylum. Thus, I would suggest either finding a source that states that most members of the phylum can induce host inflammatory responses, or scale down the comment. E.g. “members of the Fusobacteria phylum can induce”. Later (346-361) you comment on the reduced relative abundance of the genus Fusobacterium (to which F. nucleatum belongs). I think you should just discuss this here, and avoid mentioning Kostic et al when commenting on the phyla.

---

## Round 0.3 · accepted · Accept

Congratulations,your manuscript will be undergoing galleyproof processing.

·

Basic reporting

No comment

Experimental design

No comment

Validity of the findings

No comment

Additional comments

I believe that the manuscript has ended up being very good, with well presented and interesting results. I highly recommend accepting the manuscript as it now stands.